# *Yarrowia lipolytica* Adhesion and Immobilization onto Residual Plastics

**DOI:** 10.3390/polym12030649

**Published:** 2020-03-12

**Authors:** Alanna Botelho, Adrian Penha, Jully Fraga, Ana Barros-Timmons, Maria Alice Coelho, Marian Lehocky, Kateřina Štěpánková, Priscilla Amaral

**Affiliations:** 1Department of Biochemical Engineering, Escola de Química, Universidade Federal do Rio de Janeiro, R.J. 21949-900, Brasil; lanna.mbotelho@gmail.com (A.B.); acbp_adrian14@hotmail.com (A.P.); jully.lfraga@gmail.com (J.F.); alice@eq.ufrj.br (M.A.C.); 2CICECO-Aveiro Institute of Materials, Departamento de Química, Universidade de Aveiro, 3810-193 Aveiro, Portugal; anabarros@ua.pt; 3Centre of Polymer Systems, Tomas Bata University in Zlín, Tr. Tomase Bati 5678, 76001 Zlín, Czech Republic; lehocky@post.cz (M.L.); k1_stepankova@utb.cz (K.Š.)

**Keywords:** polymer surfaces, immobilization, *Yarrowia lipolytica*, recycling, adhesion

## Abstract

Research in cell adhesion has important implications in various areas, such as food processing, medicine, environmental engineering, biotechnological processes. Cell surface characterization and immobilization of microorganisms on solid surfaces can be performed by promoting cell adhesion, in a relatively simple, inexpensive, and quick manner. The adhesion of *Yarrowia lipolytica* IMUFRJ 50682 to different surfaces, especially potential residual plastics (polystyrene, poly(ethylene terephthalate), and poly(tetrafluoroethylene)), and its use as an immobilized biocatalyst were tested. *Y. lipolytica* IMUFRJ 50682 presented high adhesion to different surfaces such as poly(tetrafluoroethylene) (Teflon), polystyrene, and glass, independent of pH, and low adhesion to poly(ethylene terephthalate) (PET). The adhesion of the cells to polystyrene was probably due to hydrophobic interactions involving proteins or protein complexes. The adhesion of the cells to Teflon might be the result not only of hydrophobic interactions but also of acid–basic forces. Additionally, the present work shows that *Y. lipolytica* cell extracts previously treated by ultrasound waves (cell debris) maintained their enzymatic activity (lipase) and could be attached to polystyrene and PET and used successfully as immobilized biocatalysts in hydrolysis reactions.

## 1. Introduction

The industrial application of polymeric materials, especially plastics, has grown intensively due to their suitable mechanical, chemical, and physical properties and low cost. Several industries, such as packaging, automotive, electronic, as well as biomedical industries, use these materials for several purposes. The massive dissemination of plastics is having a negative environmental impact, mainly on the sea [1]. Several researchers have been studying the biodepolymerization of plastics [2,3], but still the reuse/recycle of those materials seems to be a simpler and more practical alternative.

Immobilizing microorganisms on solid surfaces is a relatively simple, inexpensive, quick, and hence popular process [4]. Researchers have already achieved the adhesion of microbial cells to solid polymer surfaces [5,6,7,8]. Some have tested these immobilized catalysts in important industrial reactions, for example biodiesel production [6]. For this purpose, the cells must adhere strongly to a surface to avoid desorption during the reaction with consequent loss of the catalyst [7].

The investigation of the mechanisms of cell adhesion is essential for various fields such as food processing, medicine, environmental engineering, biotechnological process in biphasic medium, etc. The adhesion of pathogens to surfaces is the primary step leading to biofilm formation and associated infections, but cell adhesion and aggregation are also widely exploited in biotechnology for immobilizing or separating microbial cells [9].

The initial steps of the cell adhesion process are suggested to be controlled by van der Waals forces and electrostatic interactions described by the classical Derjaguin–Landau–Verwey–Overbeek (DLVO) theory of colloidal stability [10]. Since long-range electrostatic repulsion is markedly influenced by the electrokinetic properties of both cells and substrate, the determination of these properties is important for a better understanding of the adhesion phenomenon. Most surfaces and microbial surfaces are negatively charged at neutral pH, and consequently, their electrostatic interactions are generally repulsive. However, interactions not included ib the DLVO model influenced by the hydrophobicity of cells and solids have been reported to also contribute to adhesion [11].

*Yarrowia lipolytica* is a unique strictly aerobic yeast with the ability to produce a wide spectrum of molecules. It is considered non-pathogenic, and several processes based on this organism were classified as GRAS (Generally Recognized as Safe) by the USA Food and Drug Administration (FDA) [12]. It is one of the most extensively studied “non-conventional” yeasts, which is currently used as a model for the study of protein secretion, dimorphism, degradation of hydrophobic substrates, and several other topics [13]. It has been previously determined by several methods of surface characterization that the cell surface of *Y. lipolytica* IMUFRJ 50682 is hydrophilic and is highly attracted to hydrophobic surfaces or molecules when previously immersed in water [14]. The adhesion of *Y. lipolytica* to Teflon-like thin films deposited on plasma-treated polycarbonate substrates resulted in a strong adhesion of cells, indicating its use for fixed bed biofilm reactors [7]. This yeast has also been studied broadly for lipase production [15,16,17]. In addition to the extracellular lipase fractions, *Y. lipolytica* can produce fractions that remain bound to the cell wall [15], as well intracellular fractions [16]. Fraga et al. [17] revealed that *Y. lipolytica* cells treated with ultrasound (cell debris) can be used as a catalyst in the hydrolysis of fat.

In recent work, cells of *Rhizopus oryzae* with a 1,3-positional specificity lipase were immobilized on biomass support particles (polyurethane foam particles) and investigated for the methanolysis of soybean oil. This process is considered to be promising for biodiesel fuel production [6]. Polystyrene and poly(ethylene terephthalate) (PET) are polymers intensively employed for food and pharmaceutical packing because of their properties. However, these versatile materials have one main drawback: they increase the amount of plastic waste [18]. The possibility of reusing/recycling these materials is of great importance to reduce their environmental impact.

The purpose of this work was to investigate the adhesion of *Y. lipolytica* IMUFRJ 50682 on surfaces with different electrokinetic properties, especially potential residual polymers (polystyrene, poly(ethylene terephthalate), and teflon), and to verify the possibility of its use as immobilization material by testing the enzymatic activity of the adherent cells, which is a crucial parameter for adequate biodepolymerization.

## 2. Materials and Methods

### 2.1. Materials

Peptone, yeast extract, and glucose were obtained from Merck (Darmstadt, Germany), Oxoid (Hampshire, UK), and Isofar (Rio de Janeiro, Brazil), respectively.

### 2.2. Adhesion Surfaces

Polystyrene, PET, and poly(tetrafluoroethylene) glass microscope coverslips (Glass) were cut to a size of 9 by 18 mm. The surfaces were cleaned by submerging in ethanol (70%, *v*/*v*) for 1 h, after which they were rinsed with deionized water and air-dried.

### 2.3. Strain, Media, and Culture Conditions

A wild-type strain of *Y. lipolytica* IMUFRJ 50682 was selected from an estuary in the vicinity of Rio de Janeiro, Brazil [19], and conserved at 4 °C on YPD-agar medium (*w*/*v*: yeast extract, 1%; peptone (from casein), 2%; glucose, 2%, agar, 3%). The cells were cultivated for 48 h at 28 °C in a rotary shaker at 160 rpm, in flasks containing YPD medium (*w*/*v*: yeast extract, 1%; peptone, 2%; glucose, 2%).

*Y. lipolytica* W29 (ATCC20460; CLIB89) was donated by the Biological Engineering Center from Universidade do Minho (Braga, Portugal). This strain was used to compare the adhesion characteristics of two different strains of the same species.

### 2.4. Adhesion Assays

#### 2.4.1. Samples Preparation

For all methods used to characterize the cell surface, the collection and preparation of samples was performed as follows: cells grown for 48 h on YPD medium were harvested (3000 g, 10 min), washed twice with distilled water, and resuspended in different buffers. This procedure was performed in order to ensure the complete removal of any substance that was not covalently linked to the cell surface, in particular, a possible surfactant.

#### 2.4.2. Adhesion Assays

The adhesion tests were based on the assays of cell adherence to polystyrene proposed by Rosenberg [20] and modified by Lehocky et al. [7] to measure hydrophobicity. For these tests, the cells were resuspended in phosphate buffer, at pH 3.0, 5.0, 7.0, or 9.0 and ionic strength varying from 10^−4^ to 10^−1^ M, until they reached an optical density at 570 nm (OD_570_) of 0.70. One milliliter of the cell suspensions was poured onto the various adhesion surfaces and left to settle for 24 h. For each sample, the supernatant was then removed by inserting the dish (10 times) in a 2 L Becker containing 1.5 L of deionized water agitated at 1000 rpm. After 2 h, the dish was observed with an Olympus optical microscope BX 51 (Olympus Europa SE & Co., Hamburg, Germany), and the images obtained were treated by an image analysis procedure to determine cell surface coverage values.

#### 2.4.3. Image Analysis

The obtained images were processed with a program developed in Matlab^®^ 6.1 (MathWorks, Natick, MA, USA) as reported by Freire et al. [21]. The image analysis performed followed a three-step sequence: image binarization, droplet quantification, and evaluation of statistical parameters. The binarization consisted in the conversion of the captured image to black and white and in noise removal. The second step quantified the cells in the image, yielding parameters such as the area occupied by the cells. During the last step, a statistical analysis of the data acquired from several images was performed in order to evaluate the average area occupied by the cells and its standard deviation.

#### 2.4.4. Pronase Treatment

Yeast cells harvested by centrifugation and washed twice were suspended in 0.01 M Tris-HCl buffer to give an OD_570_ of 10. Pronase was added to the cell suspension at a concentration of 0.1 mg mL^−1^, and the cells were harvested after incubation for a determined period of time in a shaking water bath at 37 °C [22].

#### 2.4.5. Contact Angle Measurement

Contact angles were measured by the sessile drop technique on the samples prepared previously, using an OCA 15 PLUS apparatus, Dataphysics (Filderstadt, Germany). The measurements were performed at room temperature using three different liquids with known surface tensions: water, formamide, and diiodomethane. Ten separated contact angle readings for each testing liquid were averaged to obtain representative contact angle values which were used for further evaluation according to the Young–Good–Girifalco–Fowkes equation [23]. The total surface energy (γ^tot^) and their components Lifshitz–van der Waals (γ^LW^) and Lewis acid-base (γ^AB^)—which in has a positive (γ^+^) and a negative (γ^−^) component—were calculated.

### 2.5. Testing the Immobilization of Cell Debris in Solid Surfaces

#### 2.5.1. Production of Cells with Enzymatic Activity (Cell Debris)

*Y. lipolytica* cells obtained after the fermentation process (4 L bench New Brunswick MF-114, Sci. Inc., USA, bioreator, containing 3 L of YPRFO medium *w*/*v*: yeast extract 1%; peptone, 2%; residual frying oil 2.5% *v*/*v*) were washed with distilled water and 200 mM MOPS (3-morpholinopropane-1-sulfonic acid) buffer pH 7.0 (Merk, São Paulo, Brazil) and then centrifuged at 4 °C, 4600× *g*, for 5 min. the cells were resuspended in MOPS buffer and sonicated in a 20 kHz horn-type sonicator (ultrasonic mixing sonicator, DES500, Unique Group, São Paulo, Brazil) in an ice water bath, in two stages of constant acoustic power of 150 W and frequency of 20 kHz, for 9 min. After centrifugation (4 °C, 4600× *g*, for 5 min), the sonicated biomass (cell debris with lipase) was resuspended in 200 mM MOPS buffer pH 7.0 and frozen for subsequent measurement of enzyme activity. This sonicated biomass with high lipase activity [16,17] was used as a catalyst. It was resuspended in MOPS buffer, let adhere to solid surfaces (polystyrene and PET), as described in Section 2.4.2, and used as an immobilized biocatalyst.

#### 2.5.2. Determination of Enzymatic Activity

The determination of the enzymatic activity of lipase in cell debris and cell debris immobilized on solid surfaces (polystyrene and PET) was performed by measuring the hydrolysis of *p*-nitrophenyl laurate (*p*NP-laurate) [16]. In this method, 25 mL of 560 μM *p*NP-laurate dissolved in 50 mM potassium phosphate buffer (pH 7.0) containing 1% (*v*/*v*) dimethyl sulfoxide (DMSO) was mixed, at 37 °C, either with 0.1 mL of cell debris resuspended in phosphate buffer or with the cell debris immobilized on the solid surfaces. The production of *p*-nitrophenol (product of the enzymatic reaction) was followed during 100 s in a HACH spectrophotometer, DR/4000U, (Loveland, CO, USA) at λ = 410 nm (the extinction coefficient under these conditions was 10.052 × 1/mol/cm). One lipase unit (U) is defined as the amount of enzyme which releases 1 μmol of *p*-nitrophenol per minute at pH 7.0 and 37 °C.

## 3. Results

### 3.1. Solid Surface Characterization

The characterization of the surfaces used in the adhesion tests was carried out by contact angle measurement using the sessile drop technique. The acid–base theory was used to calculate the surface energy of the samples. The total surface energy of a surface i, γ_i_^tot^, consists of an apolar, or Lifshitz–van der Waals, component, (γ_i_^LW^, which comprises the dispersion as well as the induction and orientation contributions to the van der Waals interactions) and a polar, or Lewis acid–base component (γ_i_^AB^) [24]:(1)$$\gamma_{i}^{\mathrm{tot}}=\gamma_{i}^{\mathrm{LW}}+\gamma_{i}^{\mathrm{AB}}$$

According to Lewis, the acid–base interaction can be determined by Equation (2):(2)$$\gamma_{i}^{\mathrm{AB}}=2\sqrt{\gamma_{i}^{+}\gamma_{i}^{-}}$$
where γ_i_^+^ is the electron donor, and γ_i_^−^ is the electron acceptor of the acid–base part of the surface energy.

The surface components, γ_i_^LW^, γ_i_^+^, and γ_i_^−^, can be determined by contact angle measurements, with at least three different liquids (of which two must be polar), using Young’s equation in the following form:(3)$$\left(1+\cos\theta \right)\gamma_{j}=2\left(\sqrt{\gamma_{i}^{\mathrm{LW}}\gamma_{j}^{\mathrm{LW}}}+\sqrt{\gamma_{i}^{+}\gamma_{j}^{-}}+\sqrt{\gamma_{i}^{-}\gamma_{j}^{+}}\right)$$
where j refers to the studied material, i to the testing liquid, and θ to the measured contact angle.

The liquids used in our experiments and their characteristic parameters are listed in Table 1.

The surface free energy and the corresponding contributions calculated using Equation (3) are presented, in Table 2. According to the definition of Rijnaarts et al. [26] a surface is classified as hydrophilic for 0° < θ_w_ (contact angle between water and the surface) < 20°, intermediately hydrophobic for 20° < θ_w_ < 50°, and hydrophobic for 50° > θ_w_. According to this classification, glass is hydrophilic (θ_w_ = 16.6°), and polystyrene (θ_w_ = 90.5°), Teflon (θ_w_ = 97.3°), and PET (θ_w_ = 77.8°) are hydrophobic. Although polystyrene, PET, and teflon are all classified as hydrophobic, these surfaces present different characteristics. Completely apolar compounds have no electron donor (or electron acceptor sites and thus undergo maximum hydrophobic interactions. PET, being the material with the lowest electron donor/electron acceptor characteristics, was studied in the present work (Table 2). Polystyrene presents high electron donor and acceptor surface tension components, like glass. The difference between these two surfaces is their degree of hydrophobicity.

### 3.2. Adhesion of Cells on Solid Surfaces

Images of the adhesion assays are presented in Figure 1. The image obtained from the microscope was converted to grey scale (Figure 1a) and subjected to image process analysis in Matlab^®^ 6.1 (Figure 1b). From this image some information could be obtained, such as the area (in pixels) covered by the cells. Therefore, it was possible to determine the ratio between cell-occupied area and total area, which is the parameter herein used to evaluate the adhesion of cells. Figure 1 also shows examples of the different surfaces used in this manuscript (polystyrene in Figure 1c, glass in Figure 1d, teflon in Figure 1e, and PET in Figure 1f) with adherent cells, after image process analysis.

Figure 2 shows the adhesion of *Y. lipolytica* IMUFRJ on polystyrene. It is possible to observe that the adhesion of cells to this material is relatively high at every pH and ionic strength studied, with more than 50% of surface area covered by the cells in most cases. At ionic strength of 0.1 M, the adhesion tended to decrease. Only at pH 9, the adhesion of the cells to polystyrene did not significantly vary with the ionic strength. Rijnaarts et al. [26] have shown that the isoelectric point (IEP) of a microorganism is an important parameter to predict the steric properties of cell surface polymers and their consequences for cell adhesion. The IEP of a cell surface is determined by the balance between the charges of anionic and cationic acid/base groups on the cell surface. In a previous work [14], we determined that the IEP of *Y. lipolytica* IMUFRJ 50682 is about 2.4, which indicates the presence of cell wall glucuronic acids or other polysaccharide-associated carboxyl groups. These polymers might inhibit cell adhesion at high ionic strength (0.1 M) because of steric interactions.

When glass was used as the adhesion material, the specific area occupied by the cells was smaller in comparison to the that measured when using polystyrene, though still large (around 45%), as Figure 3 depicts. It can also be observed that cell adhesion at a higher ionic strength (0.1 M) decreased. This adhesion behavior is similar to adhesion on polystyrene, which showed that at high ionic strength, adhesion was also inhibited by steric interactions. The difference between adhesion to polystyrene and adhesion to glass showed that the greater adhesion to polystyrene might be due to hydrophobic interactions between cells and this surface, as polystyrene (θ_w_ = 90.5°) is much more hydrophobic than glass (θ_w_ = 16.6°).

Another *Y. lipolytica* strain, *Y. lipolytica* W29, was also tested for adhesion. These two strains present similar surface characteristics as shown by Amaral et al. [14] and Aguedo et al. [27] (*Y. lipolytica* IMUFRJ: IEP = 2.4; θ_w_ = 28.0°; *Y. lipolytica* W29: IEP = 2.5; θ_w_ = 27.4°). Table 3 shows the results for the adhesion of *Y. lipolytica* IMUFRJ and W29 strains to polystyrene, PET, and glass at pH 3.0, 5.0, 7.0, and 9.0. It is possible to observe that the adhesion of *Y. lipolytica* W29 was weak, independent of the material or pH, in relation to that of *Y. lipolytica* IMUFRJ. This result shows that IEP and θ_w_ cannot always predict the adhesion behavior and suggests that *Y. lipolytica* IMUFRJ has some surface components that induce adhesion.

The results showed that cell adhesion of *Y. lipolytica* IMUFRJ to polystyrene was relatively high, with more than 50% of surface area covered by the cells. This characteristic seems to be unique for this strain, since reduced adhesion was observed for *Y. lipolytica* W29. In our previous study [14], we showed, by other assays, differences between the surface properties of these strains.

Table 3 also shows that adhesion to PET was inferior to adhesion to both polyethylene and glass, in most cases. This might be related to the inferior electron donor and acceptor surface tension components of PET (Table 2), which indicates that adhesion also involves polar interactions.

*Y. lipolytica* IMUFRJ was treated with pronase in order to denature proteins in the cell wall and modify its characteristics, since proteins and protein complexes (for example, mannoproteins) are usually responsible for cell interactions with organic compounds, as reported by Amaral et al. [14]. In this test, teflon was also used as an adhesion surface since it was reported that *Y. lipolytica* present high adhesion to Teflon-like films [7]. Indeed, it can be noticed in Table 4 that adhesion of *Y. lipolytica* cells to Teflon was similar to adhesion to polystyrene (for 10^−1^ M), despite the different surface characteristics of these two materials (Table 2). It is possible to observe in Table 4 that pronase treatment significantly affected the adhesion of *Y. lipolytica* IMUFRJ to polystyrene and PET and, to a lesser extent, the adhesion to Teflon. *Y. lipolytica* IMUFRJ adhesion to glass was not modified by the denaturation of surface proteins. Therefore, the yeast components responsible for adhesion to polystyrene and glass must be completely different.

The adhesion of cells to polystyrene was higher than that to other surfaces (glass and PET) and similar to that to Teflon at higher ionic strength. Polystyrene and Teflon have higher hydrophobicity (θ_w_ > 90), suggesting that the adhesion of *Y. lipolytica* cells is probably related to hydrophobic interactions. Despite the lower hydrophobicity of glass, the total surface energy and that of its components are high. The Lifshitz–Van der Waals/acid–base (LW/AB) theory was used to obtain the total surface energy γ^tot^ and that of its components, i.e., an apolar, or Lifshitz-–an der Waals component, γ_i_^LW^ (dispersion), and polar γ_i_^AB^ acid–base component [18]. Adhesion of cells to glass was inferior than to adhesion to polystyrene, but still high (34% of coverage), which might be related to the acidic component (γ^+^) of this material. It has been shown that *Y. lipolytica* IMUFRJ surface shows a more basic character (higher γ^−^) and, therefore, acid–base forces might be related to the interaction between glass and cells. The treatment of cells with pronase did not influence adhesion to glass, as it did to polystyrene. However, for Teflon, a significant interaction with cells was also observed after pronase treatment, which indicates that acid–base forces might be related to the interaction between Teflon and cells, besides hydrophobic interaction forces. Dufrêne [9] has reviewed the studies related to forces in microbial cell adhesion and reported that cell adhesion is mediated by a multitude of molecular interactions that are specific (molecular recognition between receptors and ligands) or non-specific (hydrogen bonding, hydrophobic, van der Waals, electrostatic, and macromolecular forces).

### 3.3. Application of Cells Immobilized on Polymer Surfaces

Although it has been reported that *Y. lipolytica* cells present lipase enzymatic activity [15], a higher lipase activity is detected when the cells are treated with ultrasound, which results in cell debris associated with lipase [17]. *Y. lipolytica* cell debris were tested for lipase activity (hydrolysis of *p*-nitrophenyl laurate into *p*-nitrophenol), and a positive result was obtained, as depicted in Table 5 (before adhesion). This hydrolysis activity was determined after maintaining the cell debris resuspended in MOPS buffer (pH 7) for 24 h at 25 °C, simulating the same conditions that cells are exposed to when adhesion was performed (Section 2.4.2). Lipase in *Y. lipolytica* cell debris had previously been tested for its ability to hydrolyze lipolyzed milk fat and showed good thermal stability and best reaction conditions at 37 °C and pH 7.0 [17]. After adhesion of cell debris to polystyrene or PET, these surfaces were also tested for lipase activity, with the same amount of cell debris used in suspension. Table 5 shows that hydrolytic activity was still detected when these cell debris were adherent to polystyrene or PET. The reduction in lipase activity in relation to cell debris before adhesion might be related to the fact that not all cell debris were adherent to the surfaces, as already seen in this work for cell adhesion tests. This hypothesis is supported by the fact that inferior activity was detected for cell debris adherent to PET in relation to cell debris adherent to polystyrene (Table 5), in agreement with the fact that PET showed inferior cell adhesion.

The possibility of attaching cell debris containing lipase activity to polystyrene and PET and of using this system as an immobilized biocatalyst suggest promising applications.

## 4. Conclusions

*Y. lipolytica* IMUFRJ 50682 presents high adhesion to different surfaces such as Teflon, polystyrene, and glass, independent of pH. Cell wall glucuronic acids or other polysaccharide-associated carboxyl groups might inhibit adhesion at high ionic strength (0.1 M) because of steric interactions. The adhesion to polystyrene may be due to hydrophobic interactions between cells and this surface involving proteins or proteins complexes. Cell extracts prepared using ultrasound waves (cell debris) maintained enzymatic activity (lipase), also adhered to polystyrene and PET, and were used successfully as immobilized biocatalysts in hydrolysis reactions.

## Figures and Tables

**Figure 1 polymers-12-00649-f001:**
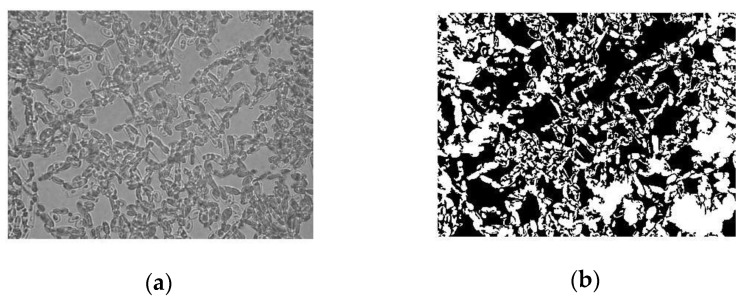
Example of the image analysis of the adhesion assays: image obtained from the microscope converted to grey scale (**a**) and after image process analysis in Matlab^®^ 6.1 (**b**). Images after image process analysis of the different surfaces tested for adhesion: polystyrene (**c**), glass (**d**), teflon (**e**), and PET (**f**).

**Figure 2 polymers-12-00649-f002:**
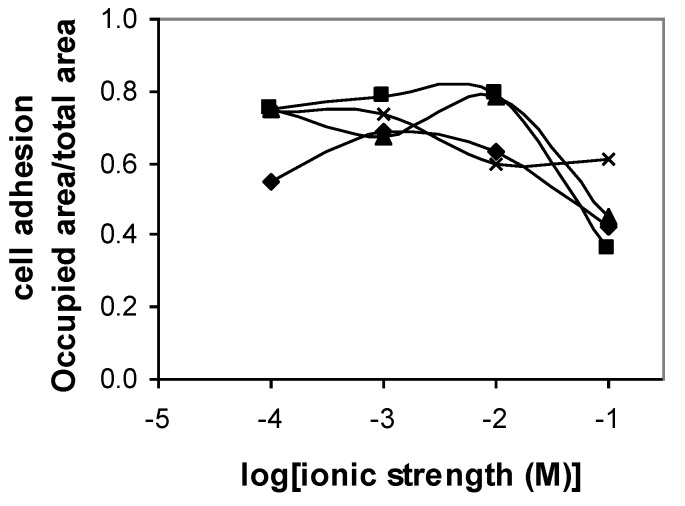
Adhesion of *Yarrowia lipolytica* IMUFRJ to polystyrene at pH 3 (♦), pH 5 (■), pH 7 (▲), and pH 9 (x) and ionic strength varying from 10^−4^ M to 10^−1^ M. The error bars are omitted for a clearer visualization. The standard deviation varied from 2% to 10%.

**Figure 3 polymers-12-00649-f003:**
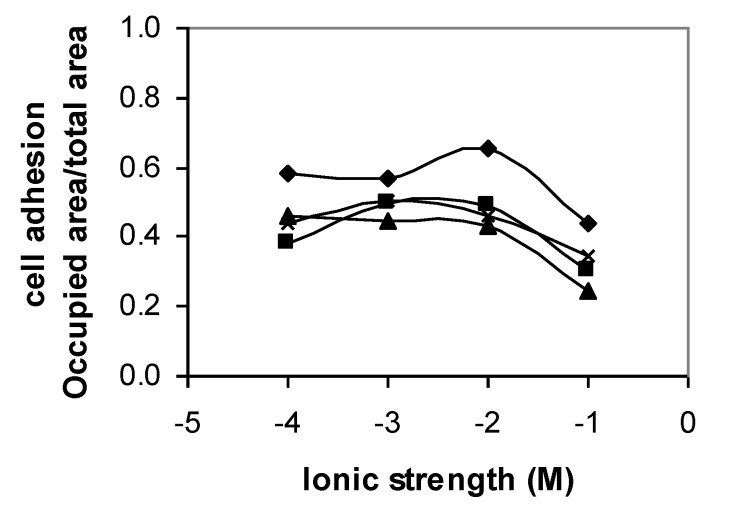
Adhesion of *Y. lipolytica* IMUFRJ to glass at pH 3 (♦), pH 5 (■), pH 7 (▲), and pH 9 (x) and ionic strenth varying from 10^−4^ M to 10^−1^ M. The error bars are omitted for a clearer visualization. The standard deviation varied from 2% to 11%.

**Table 1 polymers-12-00649-t001:** Values of the surface tension components of the testing liquids [25].

Testing Liquid	γ [mJ/m^2^]	γ^LW^ [mJ/m^2^]	γ^AB^ [mJ/m^2^]	γ^+^ [mJ/m^2^]	γ^−^ [mJ/m^2^]
Water (W)	72.8	21.8	51.0	25.5	25.5
Formamide (F)	58.0	39.0	19.0	2.28	39.6
Methylene iodide (M)	50.8	50.8	0	0	0

**Table 2 polymers-12-00649-t002:** Solid surface characteristics—contact angle, total surface free energy values—and their contributions. PET: poly(ethylene terephthalate).

Surface	Contact Angle (°)			γ^tot^	γ^LW^	γ^AB^	γ ^+^	γ^−^
	W	F	M	[mJ/m^2^]	[mJ/m^2^]	[mJ/m^2^]	[mJ/m^2^]	[mJ/m^2^]
Polystyrene	90.5 ± 0.6	27.5 ± 0.8	73.5 ± 1.1	89.8	45.2	44.6	27.1	18.3
Glass	16.6 ± 0.1	47.6 ± 0.3	51.8 ± 2.4	72.5	35.6	37.0	17.4	19.6
Teflon	97.3 ± 0.6	57.3 ± 2.9	81.3 ± 2.5	49.5	30.2	19.4	14.3	6.5
PET	77.8 ± 1.3	58.2 ± 1.8	20.5 ± 2.2	50.4	47.6	2.8	0.2	8.1

**Table 3 polymers-12-00649-t003:** Comparison of the adhesion of *Y. lipolytica* W29 and *Y. lipolytica* IMUFRJ to polystyrene, PET, and glass at ionic strength of 10^−1^ M and pH 3, pH 5, pH 7, and pH 9.

pH	Cell Adhesion Occupied Area/Total Area					
	Polystyrene		PET		Glass	
	W29	IMUFRJ	W29	IMUFRJ	W29	IMUFRJ
3.0	0.28	0.43	0.11	0.30	0.24	0.44
5.0	0.18	0.36	0.21	0.29	0.22	0.31
7.0	0.22	0.45	0.20	0.17	0.12	0.24
9.0	0.20	0.61	0.13	0.19	0.12	0.35

**Table 4 polymers-12-00649-t004:** Adhesion of *Y. lipolytica* IMUFRJ to polystyrene, PET, glass, and Teflon at pH 7 at 10^−4^ and 10^−1^ M ionic strength before and after treatment with pronase.

Material	Cell Adhesion Occupied Area/Total Area			
	10^−4^ M		10^−1^ M	
	before Pronase Treatment	after Pronase Treatment	before Pronase Treatment	after Pronase Treatment
Polystyrene	0.78 ± 0.08	0.05 ± 0.01	0.52 ± 0.06	0.07 ± 0.01
PET	-	-	0.17 ± 0.06	0.07 ± 0.04
Glass	0.34 ± 0.09	0.35 ± 0.08	0.25 ± 0.10	0.41 ± 0.05
Teflon	0.52 ± 0.07	0.39 ± 0.03	0.54 ± 0.04	0.40 ± 0.06

**Table 5 polymers-12-00649-t005:** Hydrolysis of *p*-nitrophenyl laurate by *Y. lipolytica* IMUFRJ cell debris and *Y. lipolytica* IMUFRJ cell debris adherent to polystyrene and PET.

Material	Lipase Activity (μmol *p*-Nitrophenol/min/g Cell Debris)	
	before Adhesion of Cell Debris	after Adhesion of Cell Debris
Y. lipolytica cell debris	33.66	-
Polystyrene	0	21.85 ± 6.25
PET	0	11.45 ± 1.08

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
