# Peer review of "Yarrowia lipolytica Adhesion and Immobilization onto Residual Plastics"

_polymers, 2020, doi:10.3390/polym12030649_

Round 1
Reviewer 1 Report
Comments
The manuscript entitled “Yarrowia lipolytica adhesion and immobilization onto residual plastics” reports on the adhesion of Y. lipolytica in surfaces with different electrokinetic properties and on the use of it as immobilized biocatalyst. The work described in this article is quite straightforward and easy to follow. Some minor corrections, as well as suggestions for further clarifications are outlined below:
Abstract
Line 17: no capitalisation necessary for the word “surface”
Line 24: “proteins complexes” should be corrected to “protein complexes”
Line 26: Sonication produces cell extracts and debris therefore the term "cells" should not be used here. One can use the terms "cell extracts" or "lysate".
Introduction
Line 58: The sentence “Yarrowia lipolytica, a unique strictly aerobic yeast with the ability to produce a wide spectrum of products.” is missing a verb.
Line 67: “remain bound to cell…” should be “remain bound to cell wall…”
Results
Line 194: “Adhesion of cells in the solid surfaces” should be corrected to “Adhesion of cells on solid surfaces”. Also, for this whole section, representative images showing the area occupied by cells on different substrates would be a useful addition.
Line 202: “…cell adhesion of Y. lipolytica IMUFRJ in polystyrene…” should be corrected to “…on polystyrene…”
Line 253: Before this point, there is no mention of Teflon in the text, as well as no justification of why it is also used. A short explanation should be added.
Line 255: replace “are” with “must be”
Line 260: Why was PET tested at 0.1M when all the others were tested at a lower ionic strength? This needs to be explained somehow.
Line 288: replace “cells” with “cell debris”
Lines 288-289: A simple way to test this hypothesis is to save the unbound portion of the cell debris and test it for lipase activity.
Conclusions
Lines 303-304: The sentence “Cells that have been treated with ultrasound waves…” should be replaced with "Cell extracts prepared using ultrasound waves...", as the term "cells" cannot be used after the ultrasound treatment, due to all cells being disrupted.

Reviewer 2 Report
The manuscript entitled " Yarrowia lipolytica adhesion and immobilization onto residual plastics ” focused on the adhesion characterization of Y. lipolytica IMUFRJ 50682 to different surfaces, especially potential residual plastics (polystyrene, poly(ethylene terephthalate) and poly(tetrafluoroethylene). However, I found another similar reported study “Amaral, P. F. F., Lehocky, M., Barros‐Timmons, A. M. V., Rocha‐Leão, M. H. M., Coelho, M. A. Z., & Coutinho, J. A. P. (2006). Cell surface characterization of Yarrowia lipolytica IMUFRJ 50682. Yeast, 23(12), 867-877”, which was also characterize cell surfaces—adhesion to polystyrene by the same strain IMUFRJ 50682. In addition, the last part about immobilized biocatalyst was short of the analyses of stability and catalytic parameters. So, there is not enough remarkable highlight in this manuscript.
Round 2
Reviewer 2 Report
This manuscript can be accepted.